# A Review of Research on the Use of Selected Grass Species in Removal of Heavy Metals



Tetiana Sladkovska [1,2,*], Karol Wolski [1], Henryk Bujak [3,4], Adam Radkowski [5] and Łukasz Sobol [6]

[1] Department of Agroecology and Plant Production, Wroclaw University of Environmental and Life Sciences, Grunwaldzki 24A, 50-363 Wroclaw, Poland
[2] Polissia National University, 7 Stary Bulvar St., 10008 Zhytomyr, Ukraine
[3] Department of Genetics, Wroclaw University of Environmental and Life Sciences, Grunwaldzki 24A, 50-363 Wroclaw, Poland
[4] Research Centre for Cultivar Testing, Slupia Wielka 34, 63-022 Slupia Wielka, Poland
[5] Department of Agroecology and Plant Production, University of Agriculture in Krakow, Mickiewicza 21, 31-120 Krakow, Poland
[6] Department of Applied Bioeconomy, Wroclaw University of Environmental and Life Sciences, Chelmonskiego St. 37a, 51-630 Wroclaw, Poland
[*] Correspondence: tetiana.sladkovska@upwr.edu.pl

**Abstract:** Soil and air pollution are main problems posing a serious threat to human health. Traditional physical and chemical soil remediation methods affect the soil ecosystem and are rather costly. Since the main purpose of soil remediation is not only to remove pollutants but also to restore soil health, the method of phytoremediation is becoming extremely relevant. Phytoremediation is an environmentally friendly and natural process of removing pollutants from the environment. Cleaning up contaminated sites and enabling re-use without harming future users requires the implementation of environmentally friendly and economically attractive technologies. Phytoremediation does not adversely affect the structure and biological life of the soil. Concerning on-site cleaning in situ. Hyperaccumulator plants can accumulate heavy metals from the soil, which is the so-called phytoextraction. The ability of trees and shrubs to effectively remove solid particles from the air has also been proven. However, it is not always possible to grow large plants in polluted areas. Therefore, the main goal of the research was to explore previous studies on the phytoremediation capability of herbaceous plants, in particular, their phytoextraction capacity. Another major issue was to study the main methods of improving plant phytoextraction. The results obtained show that grass can be a good solution for natural ecosystem cleanup. It is also necessary to pay attention to the impact of phytoextraction-improving substances on soil health.

**Keywords:** phytoremediation; phytoextraction; pollution; grasses; heavy metals

## 1. Introduction

Industrial activity degrades the environment with heavy metals, the quantity of which has increased significantly in recent decades. Global man-made pollution began in the middle of the last century. The situation continues to worsen as world production increases. An extremely serious problem is the elimination of toxic waste that has accumulated in the soil. Toxic elements can be found in the natural environment. Human activity is the greatest source of their spread in nature. Currently, soil plays a decisive role in food production, as well as being the most valuable ecosystem for humans [1–4]. Urban pollution is increasing due to sulfur dioxide, carbon monoxide, nitrogen oxides, carbohydrates, aldehydes, particulates, and heavy metals. These substances are toxic to animals, humans, and the environment [5]. According to Rai and Panda, air pollution not only harms human health but also changes the ecosystem as a whole, negatively affecting plants by reducing the number of photosynthetic pigments, net photosynthetic productivity,

and conductivity [6]. The World Health Organization recognizes the polluted air as a "silent killer" [7,8]. To provide mankind with clean agricultural products, it is necessary to reduce the concentration of heavy metals in agricultural soils. Cleaning up cities is equally important, as it is vital for people living in degraded areas and where human activity causes daily pollution which worsens living conditions. Excessive pollution is a serious threat to the health of the entire ecosystem [2,9]. The biggest problem related to degradation of heavy metals is that metals do not decompose and pose a chronic threat to the environment. The presence of metals such as Fe, Zn, Mn, Cu, Ni, and Co in small amounts is necessary for the proper growth and development of plants, but their excessive amounts are extremely dangerous for humans, animals, and plants. However, heavy metals such as Cd, Pb, As, and Hg have important functions in plants and are phytotoxic. They can seriously interfere with physiological and biochemical processes in plants. Heavy metals and chemical pollutants that accumulate and are transported in the food chain (soil–plant–animal–human) affect various organs of animals and humans, causing diseases [10].

Physical and chemical cleaning of heavy metals from soils is a time-consuming and extremely costly process that is not always effective. It also leads to a significant generation of secondary waste [11–13]. Phytoremediation is an alternative solution to this problem, as soil remediation is performed naturally using plants. Phytoremediation is the removal or stabilization of pollutants through physiological processes in plants. In addition, this technology is much cheaper and more profitable compared to other methods [14,15]. Phytoremediation is one of the most promising and environmentally friendly methods of rehabilitation, and this method has become particularly widespread in the last twenty years [12]. Phytoremediation is also known as enhanced bioremediation because bioremediation only uses microorganisms to cleanse the soil, while phytoremediation uses a combination of plants and related microorganisms to cleanse and restore soil biology. While microorganisms can transform metals, they cannot remove them from contaminated soil. The phytoremediation mechanisms are quite complex and are not limited to the direct metabolism of pollutants [16–19]. There are some indirect mechanisms of pollution reduction, such as the action of certain root-related microbes. Transport of pollutants to the above-ground parts of the plant takes place through the absorption of water by the roots, and then transport through the xylem [20]. For example, when plants absorb heavy metals from the soil and transfer them to land parts, the process is called phytoextraction. In this way, the soil is usually cleaned of heavy metals, radionuclides, and some organic compounds. Plants used for this purpose should usually be characterized by fast growth, a branched root system, and high biomass productivity [21]. Plants can also reduce the mobility of heavy metals in the soil, a process known as phytostabilization. In this case, the pollutants may be immobilized from the soil to the surface of the roots or be in the rhizosphere of the plant [17,22]. The soils are constantly contaminated with toxic substances, including pesticide residues and heavy metals, and the surface of such land is increasing. This is due not only to the activities of heavy metallurgy and the chemical industry but also to the irrational use of chemical plant protection products in agriculture [23].

## 2. Phytoremediation and Its Main Strategies

The term "phytoremediation" was coined by Ilya Raskin, professor at the Center for Biotechnology in Agriculture and Nature Management at Rutgers University (USA) in 1989 [15,24]. Phytoremediation technology is based on the ability of plants to remove toxic substances from the environment or transform them into safe compounds—metabolites. It is a natural process carried out by plants to purify and stabilize pollutants in the environment [22]. Several phytoremediation methods are described below and shown in Figure 1.

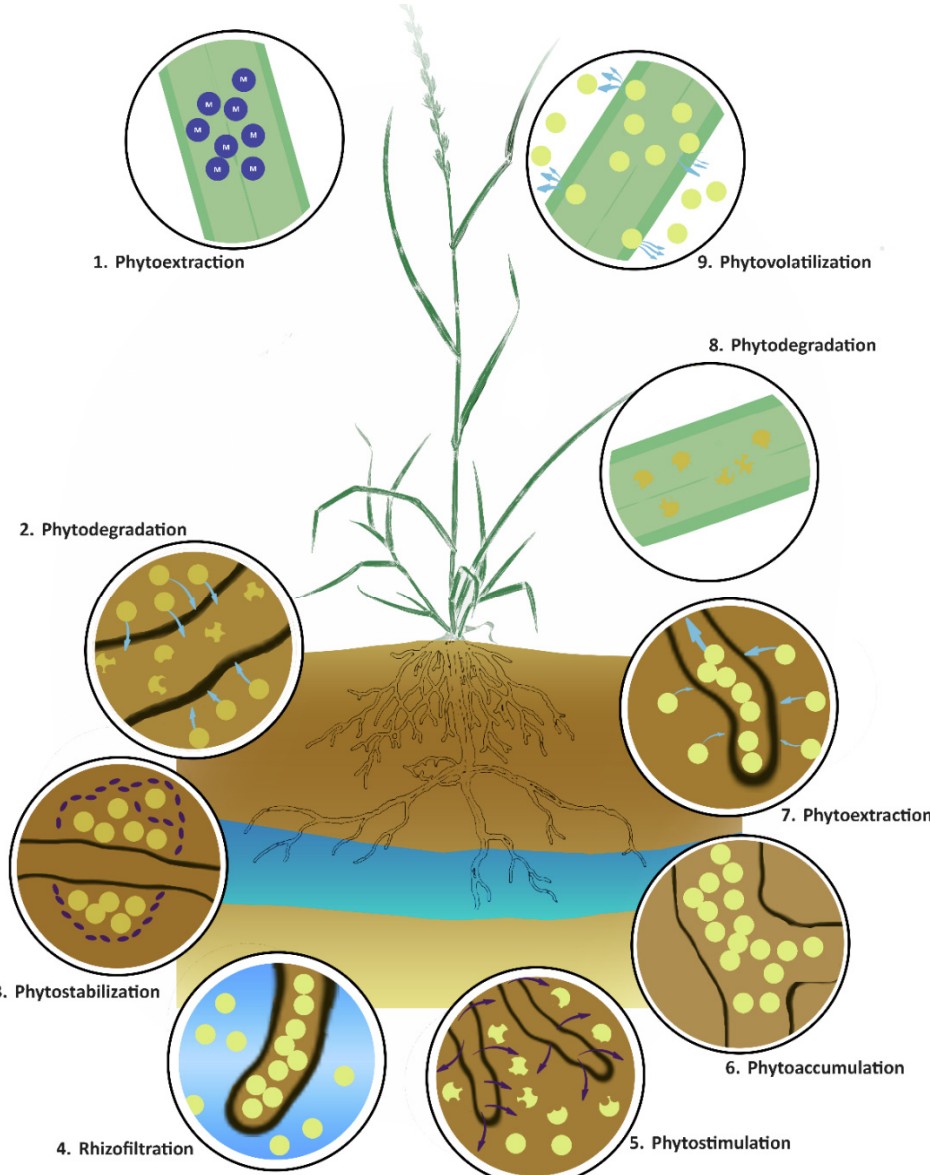

**Figure 1.** Phytoremediation strategies with the use of grasses.

- Phytoextraction is the process of accumulation or absorption by plants of environmental pollutants that accumulate in the plant but do not decompose. It is most often used to clean soil contaminated with heavy metals. The depth of soil cleaning is limited by the depth of the plant's root system. Phytoextraction can remove metals such as Cr, Cd, Cu, Co, Ag, Zn, Ni, Mo, Pb, and Hg [22,25,26]. This method, unlike phytostabilization, removes contaminants from the soil and not only stabilizes it. This process includes such steps as mobilization of heavy metals in the rhizosphere, penetration of heavy metals into plant roots, absorption of heavy metals by the roots, their movement from the roots to the above-ground parts, and the sequestration and compartmentation of heavy metal ions in plant tissues. The key factors in phytoextraction are the correct selection of the plant, and for this, it is necessary to determine the phytoextraction potential, the accumulation capacity of metals, and the above-ground biomass of the species; therefore, there are phytoextraction techniques with a lower capacity for accumulation of metals [3,27–29].
- Phytostabilization is the immobilization of pollutants in the soil through the accumulation and absorption of heavy metals by the roots, where they transform into a non-toxic

form. This method minimizes the leaching of pollutants by increasing the evapotranspiration of the system [29–31]. Plant growth restores soil activity. The advantage of this strategy is that it does not require the disposal of hazardous biomass [32]. The decisive point here is the choice of plants, as the roots play a major role in the immobilization of heavy metals and the stabilization of the soil structure [28,33–36].

- Phytodegradation is the process of decomposition of harmful pollutants in plant tissues by plant enzymes, as well as endocytic bacteria inhabiting internal plant tissues. Decreasing the concentration of metals may also occur outside the plant, due to the release of plant root enzymes, such as nitroreductase, oxidase, phosphatase, and nitrilase [37–39]. Phytodegradation is based on the secretion of natural substances by plants in the root zone, which in turn creates a breeding ground for microorganisms in the soil. These microorganisms initiate the natural degradation of toxic compounds, especially organic ones.
- Phytovolatilization is the process by which plants absorb pollutants from the soil, convert them into volatile particles, and then release these volatile chemicals into the atmosphere. The advantage of this method is that, unlike phytoextraction, phytovolatilization does not require the removal of contaminated plant organs. This method is quite effective in removing mercury [29,40]. The disadvantage of this strategy, however, is that it does not completely emit pollutants, but only transfers them from the soil to the atmosphere, where they can pollute the air or return to the soil along with rainfall [3,41,42].
- Rhizofiltration is a method in which the roots of plants absorb and collect pollutants from surface or groundwater, and this method is also suitable for coastal areas. In the process of rhizofiltration, it is possible to isolate Pb, Cd, Cu, Ni, Zn, and Cr from the environment. Once saturated with pollutants, such plants should be harvested [25,43].
- Phytoaccumulation is defined as the increase in microbial activity to degrade organic compounds by exudates from plant roots [44].

Plants can increase the bioavailability of contaminants by releasing various root exudates that alter the rhizosphere pH and increase the solubility of heavy metals. This is followed by the sorption of the metal by the root and its further movement through the cells. Absorption and mobility are due to a variety of molecules including metal ion transporters and complexing agents. These special transporters or carrier proteins are found in the cell membrane of the root cells [3,45].

During evolution, plants developed two strategies for protection against the toxic effects of heavy metals—avoidance and tolerance.

The avoidance strategy is the ability of plants to limit the uptake of heavy metals and limit their movement through the root tissues. By isolating various root exudates, plants form stable heavy metal complexes in the rhizosphere. For example, some exudates can alter the pH of the rhizosphere, leading to the precipitation of heavy metals, which in turn reduces their bioavailability and toxicity. Arbuscular mycorrhiza can block the penetration of heavy metals into the root by their absorption, adsorption, or chelation of heavy metals in the rhizosphere. In addition, the cell wall of root tissues can deposit heavy metals that are embedded in it so that they cannot move further [3,46].

A tolerance strategy is to detoxify heavy metals to minimize their impact on the plant. This is achieved mainly by chelation by complexing heavy metal ions with ligands. This reduces the concentration of free heavy metal ions to a relatively low level. Organic acids, amino acids, phytochelatins, metallothioneins, and proteins are involved in this process [3,47].

Following chelation, heavy-metal–ligand complexes are transported to inactive compartments such as vacuoles. They can also be distributed to other places that do not harm the plant, such as petioles, leaf scales, and trichomes. Heavy metals can also be transferred to old leaves and removed by their natural shedding [3,48].

### 3. Plants and Phytoremediation

Plants penetrate their root system into the soil matrix and create a rhizosphere ecosystem in which heavy metals accumulate and change their bioavailability, thus recreating the contaminated soil. In recent years, many studies have been carried out to understand the molecular mechanisms underlying plant tolerance to heavy metals, which will help in developing methods to improve the effectiveness of phytoremediation [3].

Compared to the microorganisms used for bioremediation, plants are more resistant to contamination with heavy metals and are also able to extract heavy metals from the soil and stabilize them through root sorption. Plants that have mechanisms of existence in high concentrations of metals are called metallophytes. Based on the physiology of such plants, they can be classified according to such properties as indicators: regulate the absorption of metals so that the internal concentrations reflect the external concentrations of the soil; bacteria show active metal absorption and movement to the aerial parts, except for some substances which reduce the penetration of metals into the root and/or their transport to the shoots. Some species of metallophyte plants are hyperaccumulators, and, as already mentioned, they can accumulate metals above 1% of their dry weight. For a long time, this ability of plants was considered dangerous because it could cause contaminants to enter the food chain. However, over time this property has been assessed and hyperaccumulators have become a promising biological source for phytoremediation [46–48].

Almost 600 plant species can accumulate environmental pollution: in phytoremediation, the so-called hyperaccumulator plants are used. Their feature is that they can absorb large amounts of impurities and quickly and efficiently transfer them from the roots to the shoots. They also effectively bind toxic metals into various chemical compositions, thus reducing the toxicity of these metals [2,49,50].

Hyperaccumulator plants must meet the following criteria: the ratio of heavy metal concentration from the shoot to the root is greater than 1 (the ability to transport heavy metals to the shoots), and the ratio of heavy metals between the shoot and the soil is greater than 1 (the ability to transport is absorbed by heavy metals from the soil) [3,51]. Additionally, the metal concentration in the shoot is higher than 10 mg/kg for Hg, 100 mg/kg for Cd and Se, 1000 mg/kg for Co, Cu, Cr, Ni, and Pb, and 10,000 mg/kg for Zn and Mn [52]. Currently, there are about 450 plant species from 45 angiosperm families that can be considered hyperaccumulators [3,52].

The topic of the use of trees and shrubs in phytoremediation is quite well known. However, very little research has been conducted to investigate the accumulation of pollutants in herbaceous plants, especially cereals found in urban meadows. Such meadows, unlike trees and shrubs, may be, for example, near roads, which are one of the main sources of pollution in cities. In addition, trees cannot be planted in many parts of urban areas for safety reasons. Trees growing at intersections and roadsides reduce visibility and can be dangerous to traffic. The area where trees should not be planted increases year by year with the number of roads and buildings, and most of this area is covered by intensively tended lawns [53].

According to scientists, organic compounds released in the rhizosphere of perennial grasses contribute to the development of microbial concentrations that accumulate some heavy metals. The cumulative effect, however, depends on the individual development of plants in urban pastures and the concentration of industrial pollutants in the soil [54,55]. Perennial grasses also can accumulate large amounts of heavy metals in roots and rhizomes; they are an important element of urban landscape ecosystems for building parks and creating roadside buffer greenery, green sports fields, and more. After mowing, they form a dense plant cover, have a high anti-erosion potential, and their land biomass after drying is less flammable compared to other plants used for landscaping. They improve the microclimate, promote the absorption of carbon dioxide, increase biodiversity, and improve fertility [56,57].

According to the research of Barrutia et al. (2011) on soils contaminated for 30 years with Pb, Zn, and, to a lesser extent, Cd, the area was revitalized and 31 plant species

belonging to 28 genera and 15 families were identified. The most common was *Festuca rubra* from *Thlaspi caerulescens*. Often this consortium was accompanied by *Jasione montana* and/or *Rumex acetosa*, and less often *Plantago lanceolata*. In the study area, where the impact of mining was significantly lower, the dominant species were *Ulex europaeus*, *Pteridium aquilinum*, and *Molinia caerulea* [58].

From the *Poaceae* family, the genus *Lolium* is one of the most important types of grass used in the world for forage and ornamental purposes. It is a hyperaccumulator and can accumulate 1–2% of potentially toxic elements [59,60]. *Lolium perenne* L. is a promising plant for heavy metal monitoring and regeneration due to its rapid growth, root development, metal sensitivity, and soil degradation adaptability. Leudo describes this plant in his research as hyperaccumulators [31,61]. *Lolium perenne* L. is very widely used in urban landscape architecture, but there is little research into its heavy metal toxicity and phytoremediation ability, although some scientists still refer to it as a hyperaccumulator [62,63]. The results of the Cruz study showed that *L. perenne* is a plant with significant phytoremediator capacity against heavy metals Cd and Hg, and the study showed the presence of tolerance mechanisms observed during germination and growth of roots and stems. Moreover, these studies have shown that the phytoremediation potential may be influenced by the presence of mixed metals in the environment. Regarding the morphological response of individual metals, it was found that sprouting showed an inhibitory response to increasing concentrations. However, the plant grew normally and achieved a sufficient germination percentage [62,63].

*Festuca rubra* L., a perennial grass species widespread throughout the world, grows in a variety of environmental conditions. This grass is grown in contaminated areas due to its extensive root system, good crop rejuvenation, fast growth, high biomass, and high tolerance to adverse environmental conditions. *F. rubra* has a high phytoremediation potential for contaminants such as, B, Cu, Zn, Mn, and Mo. It plays an important role in the functioning of the ecosystem, providing an excellent opportunity to introduce this type of plant to contaminated areas [31,64,65].

*Poa pratensis* L. is characterized by very intensive tillering and forms a compact turf with branching and loosely clumped structure. Early plant growth and resistance to mechanical damage, compared to other grass species, contribute to the development of this plant due to the formation of dense turf on newly developed surfaces, especially on fertile, fairly moist soils [66–68]. In his research, Wang described this species as highly tolerant to Cd, and claimed that it is a potential phytoremediation material for soil contaminated with Cd [67]. Among the studied species of grass, *Poa pratensis* seeds were the most resistant to the toxic effects of copper and were germinated under the conditions of increased concentration of this element in the soil [68,69]. In the research of Bidar et al. it was found that *L. perenne* and *T. repens* species can form a good plant cover on soil contaminated with Cd, Pb, and Zn. It was also shown that the greatest accumulation of heavy metals was observed in the roots and shoots of plants, while in the shoots of *T. repens* they were smaller compared to *L. perenne*. In turn, *L. perenne* was more exposed to oxidative stress induced by heavy metals than *T. repens* [70]. Table 1 shows the plant species, type of phytoremediation process, and bound metals based on a literature review.

**Table 1.** Examples of plant species used or the remediation of metals.

| Plant Species | Process | Metals | References |
|---|---|---|---|
| *Agrostis capillaris* | Phytostabilization | As, Pb, Cu, Ni | [35] |
| *Agrostis stolonifera* | Phytostabilization | Cd, Pb, Zn, As, Cu | [30,52] |
| *Arabidopsis halleri* | Phytoextraction | Zn, Cd | [2,15] |
| *Arabidopsis thaliana* | Phytoextraction | Cd | [12,41,50] |
| *Festuca rubra* | Phytostabilization | Zn, Cd, Pb, Cu | [31,64] |
| *Dactylis glomerata* | Phytostabilization | Cd, Zn, Pb | [31] |

**Table 1.** *Cont.*

| Plant Species | Process | Metals | References |
|---|---|---|---|
| *Festuca arudinacea* | Phytoextraction | Zn, Pb | [60,65] |
| *Lolium perenne* | Phytoextraction | Cd, Pb, Zn | [31,61,62,65] |
| *Lolium italicum* | Phytostabilization | Zn, Pb | [31,60] |
| *Melilotus officinalis* | Phytostabilization | Zn, Pb, Cu | [45] |
| *Paspalum atratum* | Phytoextraction | Zn, Cd | [31,36] |
| *Poa pratensis* | Phytoextraction | Cd, Cu | [67,68] |
| | Phytostabilization | Zn | [34] |
| *Trifolium olexandrinum* | Phytoextraction | Zn, Cd, Pb, Cu | [71] |
| *Trifolium repens* | Phytostabilization | Cd, Pb | [72] |

## 4. Methods of Increasing the Effectiveness of Phytoremediation

It is not uncommon for plants used in phytoremediation to have some limitations. They are usually slow growing and have problems with adapting to different growing conditions, and the selection and genetic engineering are used to overcome these barriers. While there are reports of increasing the effectiveness of phytoremediation to remove contaminants in agriculture, more research is still needed to extend this phenomenon.

Genetic engineering is a promising technique to increase the effectiveness of phytoremediation. Compared to the traditional selection, this method is much faster, and it is also possible to use it to add hyperaccumulation genes to plants in terms of other parameters [72]. However, there are also disadvantages, as plant detoxification mechanisms are very complex and involve many genes, so it is necessary to genetically manipulate several genes to obtain the desired result, which takes a long time and rarely gives a favorable result. Recent studies, including the overexpression of genes whose protein products are involved in the capture, transport, and sequestration of metals or act as enzymes involved in the degradation of hazardous organic substances, have opened up new possibilities in the field of phytoremediation [73]. There is also another problem with genetically modified plants: the problem with obtaining permits for conducting field experiments with such plants, which is related to the safety of ecosystems [3].

The use of plant-related microorganisms is an effective way to increase the phytoremediation capacity of plants. The use of rhizosphere microorganisms can directly stimulate plant growth and their resistance to heavy metals, as well as regulate the delivery of heavy metals to plant roots [74,75].

There are approximately 240 species of arbuscular mycorrhizal fungi (AMF) that are found in nearly all agricultural and natural ecosystems and coexist with approximately 90% of terrestrial plants. They form a direct connection between plant roots and the substrate [1]. AMF increase the absorption area and allow plants to obtain the necessary nutrients with the help of extra-spore mycelium that spreads in the soil. It dissolves and helps plants absorb nutrients such as phosphorus, nitrogen, and zinc [76]. At the same time, AMF obtain carbohydrates and lipids from the plant. This symbiosis helps make the plant more stable under stressful conditions. AMF activate antioxidant mechanisms in plants, and the mycelium begins to secrete a special glycoprotein that improves soil conditions [1,77].

According to Parvin et al., AMF have endured high levels of metal contamination, which in turn helps plants immobilize, convert, detoxify, and remove heavy metals. Therefore, it is very important to increase plant resistance to phytoremediation by inoculating them with mycorrhiza. This leads to an increase in the absorption of heavy metals by the roots, an increase in the binding of pollutants in the root and soil, and a decrease in metal transfer to the above-ground parts of plants by transforming their ions into forms with poor bioavailability [78]. In Zhang's experiments, AMF were found to increase plants' resistance to Cd toxicity by improving photosynthesis, stimulating the release of antioxidant enzymes,

and assisting in water access and nutrient uptake. AMF also help to increase the growth and biomass of the host plants by increasing the size of the roots [79]. In addition, AMF hyphae entangle metal ions above soil particles, which reduces their availability and migration due to adsorption and stabilization [1,80]. Depending on the type of plant and the AMF, the contaminants in the soil or the root of the host plant may stabilize or be transferred to the shoots.

While investigating the mechanisms of Pb accumulation in the mycelium of AMF, Salazar et al. showed that there are some mechanisms of accumulation of this metal in mycelial spores, and the amount of absorption depends on the AMF species and the host plant; the Glomeraceae species is the most important AMF species in lead-contaminated soils [81]. Moreover, AMF help to prevent the transfer of Pb to plant shoots, contributing to its accumulation in the root, and also secrete a general protein related to glomalin, which increases Pb retention in soil and also reduces its bioavailability [1].

One important strategy to improve phytoextraction efficiency is to increase the bioavailability of heavy metals. According to bioavailability, heavy metals in soil can be classified as readily available—Cu, Cd, As, Ni, Zn, and Se; moderately available—Fe, Mn, and Co; or difficult to obtain—Pb and Cr [82]. The bioavailability of metals is extremely important for the efficiency of phytoextraction. In soil, it depends on the internal availability of metals, the properties of the soil, and the binding of metal particles to the soil. This indicator is influenced by the physical and chemical composition of the soil, soil pH, microbial activity, and the presence of chelating agents. Various plant strategies exist to increase the bioavailability of metals. These are root exudates that lower the soil pH, which contributes to the formation of free heavy metal ions from insoluble complexes. Plants can also secrete phytosiderophores, organic acids, and carboxylates, which contribute to the chelation of heavy metals [75].

According to Sheoran et al., rhizosphere microorganisms are able to secrete enzymes, which significantly increases the availability of heavy metals [83]. Some endophytes secrete biosurfactants and siderophores that mobilize heavy metals in the soil. Siderophores increase their bioavailability for rhizobacteria and plants through chelation with heavy metals [75].

Chelating agents increase the desorption of heavy metals from the soil and also prevent their sorption and precipitation in the soil, which in turn increases the bioavailability of metals. Various chelating agents, including organic and synthetic, are used in chelation to improve the efficiency of phytoextraction. Synthetic chelating agents, according to Gupta et al., can effectively increase the bioavailability of heavy metals: they are ethylenediaminetetraacetic acid (EDTA), diethylenetriaminepentaacetic acid (DTPA), and ethylene glycol tetraacetic acid (AGTA) [84]. However, they do not break down well, which can lead to their accumulation in the soil. Their alternatives are organic chelating agents—oxalic, malic, and citric acid. They are of natural origin and degrade easily in the soil, which is less hazardous to the environment [85].

Currently, another type of phytoremediation is emerging, namely nanobioremediation. This method combines bioremediation and nanotechnology. Pollutants are adsorbed, modified, or destroyed thanks to the properties of nanoparticles, soil microflora, and plants. Nanoparticles can also act as catalysts and help to activate pollutant degradation [86]. According to Misra et al., the use of nanoparticles in combination with microorganisms is effective, as they accelerate the elimination of heavy metals by acting as microbial nanocarriers or microbial biosorbents [87].

Modern soil remediation methods combine biological and nanotechnological remediation. Nanoparticles can absorb a large number of different pollutants, as well as catalyze reactions that reduce the energy required for their cleavage. They can also remove pollutants by adsorption [86].

The studies of many scientists show that foliar application of nanoparticles leads to a reduction in the toxicity of heavy metals and improves plant growth, which in turn

increases the effectiveness of phytoremediation [88]. Recently, the use of carbon- and metal-based nanoparticles has become widespread.

CNT carbon nanotubes, nano-zero-value iron (nZVI), TiO$_2$NPs, and AgNPs are mainly used for the remediation of soil contaminated with heavy metals, in addition to heavy metal removal efficiency according to pH and ambient temperature, contact time, and dose [89].

Despite the many positive aspects of the use of nanotechnology in phytoremediation, there is concern about the release of these substances into the environment and their subsequent toxicological effects. The use of nanoparticles can lead to their accumulation in soil, which in turn will affect soil properties. According to the research by Rajput et al., the presence of nanoparticles can change soil pH, which is one of the most important parameters and depends on the availability of nutrients, microbiological dynamics, the general condition of the soil, and plant growth and development [90]. According to recent studies, the use of nZVI can significantly increase the pH value of a soil solution. Ag, Au, Ti, and Zn nanoparticles, in addition to affecting soil pH, also have a negative effect on soil microorganisms and nematodes. Their degree of harmfulness is influenced by their enzymatic activity, concentration, and type. An increase in the concentration of nanoparticles is also associated with a decrease in dehydrogenase, which in turn disturbs the balance of nutrients and soil fertility, affects the mycelium, and disrupts the functioning of microbial cells [86,91–93].

When nanoparticles enter water bodies, they become pollutants and are absorbed by aquatic organisms and become toxic to them, as they influence biochemical processes at the molecular and tissue-organic levels. The most common toxic effects caused by nanoparticles are DNA damage, oxidative stress, clotting disorders, and cell death, and they can also cause immunomodulatory and immunosuppressive effects in biological systems [86,94–96]. Currently, nanoparticles can be found in water, soil, and air, and their widespread use will lead to their spreading; therefore, they can penetrate into food chains and be toxic to animals and humans [97–99].

## 5. Conclusions

Species with significant phytoremediation capacity are *Lolium perenne*, *Festuca rubra*, and *Poa pratensis*. These species belong to the basic species in the composition of lawn mixtures. From a broader environmental perspective, it is important to recommend other species as well. Namely, several grass species can be used for phytoextraction of slightly contaminated agricultural soils or for phytostabilization of soils with a low or moderate concentration of trace elements; only a few species are suitable for phytofiltration of trace elements. Grasses have great potential to stabilize trace elements in soil, sediment, and wastewater.

The use of grasses for phytoremediation is limited to the topsoil. Rooting depth varies depending on the type of grass used, and often most of the root system is located in the upper part of the soil, 0–15 cm. Grasses often give a high yield of biomass (sometimes over 20 $t_{DM} \cdot ha^{-1} \cdot year^{-1}$), which can be used after harvest as a source of bioenergy. Another advantage is the fact that perennial grasses that are exposed to trace elements can regrow (grow) after being mowed.

Currently, more detailed research using molecular approaches is needed to better understand the processes involved in the uptake, transport, and accumulation of trace elements. Research should be carried out to identify trace element transporters and elucidate the mechanisms related to the uptake, transport, and accumulation of trace elements, especially for Cr, Hg, Ni, and Pb.

Tests must be carried out under more realistic conditions, as most grasses die (stop growing) after high doses of various trace elements are applied. The effectiveness of selected grass species in phytoextraction or phytostabilization can be improved with plant growth promoting microorganisms, but the adoption of such a strategy should be assessed on a case-by-case basis. There may be an increase in the uptake of trace elements by plants

instead of better phytostabilization of trace elements and vice versa due to the specificity of the trace elements, patch/microorganisms, and plant interactions.

In general, remediating sites contaminated with trace elements using only grasses has proven to be a successful strategy [100,101], but growing grasses associated with other crops such as legumes and trees can speed up land reclamation and it is more ecologically adequate [102,103].

**Author Contributions:** Conceptualization, T.S. and K.W.; methodology, K.W.; software, A.R., H.B. and Ł.S.; validation, T.S., K.W. and A.R.; formal analysis, K.W.; investigation, T.S. and K.W.; resources, Ł.S.; data curation, Ł.S.; writing—original draft preparation, T.S.; writing—review and editing, Ł.S. and A.R.; visualization, T.S.; supervision, K.W.; project administration, T.S. and K.W. All authors have read and agreed to the published version of the manuscript.

**Funding:** This research received no external funding.

**Institutional Review Board Statement:** Not applicable.

**Informed Consent Statement:** Not applicable.

**Data Availability Statement:** Not applicable.

**Conflicts of Interest:** The authors declare no conflict of interest.

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
