# Peer review of "A Review of Research on the Use of Selected Grass Species in Removal of Heavy Metals"

_agronomy, doi:10.3390/agronomy12102587_

Round 1

Reviewer 1 Report

The authors of  " The Review of Research on the Use  of Grasses in Removing Pollutants" provide an overview about the use of grasses for phytoremediation and knowledge of the underlying mechanisms. The topic of the review is of high relevance and suitable to be published in agronomys in general. Since the removing pollutants traditional physical and chemical methods both affect the soil ecosystem and rather costly, I think that the topic is very interesting because it covers an important aspect that to explore previous studies on the phytoremediation capability of herbaceous plants, in particular, their phytoextraction capacity. An extensive review of literature regarding this topic was performed by the authors. However, from my point of view there are considerable flaws in the manuscript which need major revisions. The manuscript does not provide a critical discussion of contradictory results in the literature or a kind of a bigger pictureurse, this is challenging because of the complexity of phytoremediation and soil, the  gaps both in knowledge and practice. But in its present state, the manuscript is more a collection of existing studies than a comprehensive review providing a systematic view on the topic.

Also I think the authors have to improve their article with the following suggestions:

1)       The title of this review article indicates the usage of grasses to remediate pollutants, while the content summarized several ornamental grass species with the ability of removing heavy metals. There are more up-to-date studies on grasses removing organic or inorganic contaminants both on land and in water. The author may either adjust the title of the article or including more relevant research results into the review.

2)       The abstract mentions that is a review about soil phytoremediation which not only to remove pollutants but also to restore soil health; then what is missing in your article is a paragraph in which you describe better the link between remove pollutants and restore soil health. So it is necessary that you reinforce this link and help to understand why remove pollutants led to soil health functions and development.

3)      Line 42- 44, here the author mentioned several pollutants that cause urban pollution, is there a better way to classify various contaminants and won’t overlap with each other.

4Line 68-70, the former sentence is in conflict with the later.

5  Line 101-102, the phytoremediation strategies illustrated in the graph did not fully match the following text contents. It is better to list out the definitions of those strategies.

6  Line 113-117, are xylem and root system relevant to the phytoextraction potential? For example, does the length of the roots indicate the limitation of certain repairing plant?

7Line 118-125, does phytostabilization happen at above ground parts of a plant, such as the leaves adsorb particulates from air?

8)      Line 156, “Arbuscular mycorrhiza” should be italic.

9)      Line 189-199, the content here are more or less repeating the phytoremediation strategies.

10Line 280, there are plenty studies about intercropping enhancing the effectiveness of phytoremediation, and it’s effective way to removing pollutants in agriculture.

11 The manuscript has done a good work in summary of previous literatures but "outlook" is lacking, which should draw attention and thinking from the authors I think. You should expand the recommendations, i.g., phytoremediation with hyperaccumulation rice varieties, both farmland pollution repair and crop production at the same time, some references added should be beneficial for the article.

Reviewer 2 Report

This manuscript reviewed phytoextraction of pollutants with herbaceous plants. Overall, this manuscript is well organized, and might be interesting and suitable for the readers of this journal. I suggest it could be accepted for publication after minor revision. 

1. The “Pollutants” in the title was suggested to be replaced by “Heavy Metals” or “Toxic Metals”.

2.Line 24, “As a rule, plants having massive herbage, the so-called hyperaccumulators,” the concept of hyperaccumulator here was not precisely correct.

3.Part 2, “Phytoremediation and its Main Strategies”, could have been shorten. Part 3, “Plants and Phytoremediation”, could the authors reorganize the part or add some more information? For example, classify the plants, or mention some phytoextraction mechanisms that those grasses use.

Reviewer 3 Report

Reviewer Comment to manuscript agronomy-1957873

[The Review of Research on The Use of Grasses in Removing Pollutants]

The article is publishable after clarifying the comments:

Review article prepared correctly. It contains the appropriate number of bibliography items that have been correctly cited in its content. The subject of the work is interesting and can help in practical activities. However, this is not a fully innovative study, because similar studies have already been published, for example:

Rabêlo FHS, Vangronsveld J, Baker AJM, van der Ent A and Alleoni LRF (2021) Are Grasses Really Useful for the Phytoremediation of Potentially Toxic Trace Elements? A Review. Front. Plant Sci. 12:778275. doi: 10.3389/fpls.2021.778275

Therefore, I am asking the authors to provide justification, which is innovative in their review article.

And one more little note:

Line 45; “According to Rai & Panda…” – no bibliography was given.

.

Round 2

Reviewer 1 Report

The authors have revised their manuscript following the reviewer suggestions. There are no major problems to be addressed. However, I think the comments and suggestions for Authors in Review Report 1  still be available and would be benefit to the manuscript